# Dysregulation of Macropinocytosis Processes in Glioblastomas May Be Exploited to Increase Intracellular Anti-Cancer Drug Levels: The Example of Temozolomide

**DOI:** 10.3390/cancers11030411

**Published:** 2019-03-22

**Authors:** Margaux Colin, Cédric Delporte, Rekin’s Janky, Anne-Sophie Lechon, Gwendoline Renard, Pierre Van Antwerpen, William A. Maltese, Véronique Mathieu

**Affiliations:** 1Department of Pharmacotherapy and Pharmaceuticals, Faculty of Pharmacy, Université Libre de Bruxelles (ULB), 1050 Brussels, Belgium; Margaux.Colin@ulb.ac.be or macolin@ulb.ac.be (M.C.); alechon@ulb.ac.be (A.-S.L.); renardgwendo@hotmail.com (G.R.); 2RD3-Pharmacognosy, Bioanalysis and Drug Discovery Unit and Analytical Platform, Faculty of Pharmacy, Université libre de Bruxelles (ULB), 1050 Brussels, Belgium; Cedric.Delporte@ulb.ac.be (C.D.); pvantwer@ulb.ac.be (P.V.A.); 3VIB Nucleomics Core, VIB, 3000 Leuven, Belgium; Rekins.Janky@vib.be; 4Department of Cancer Biology, University of Toledo College of Medicine, Toledo, OH 43614, USA; william.maltese@utoledo.edu; 5ULB Cancer Research Center, Université libre de Bruxelles (ULB), 1050 Bruxelles, Belgium

**Keywords:** glioma, macropinocytosis, methuosis, honokiol, vacquinol-1, MOMIPP, temozolomide

## Abstract

Macropinocytosis is a clathrin-independent endocytosis of extracellular fluid that may contribute to cancer aggressiveness through nutrient supply, recycling of plasma membrane and receptors, and exosome internalization. Macropinocytosis may be notably triggered by epidermal growth factor receptor (EGFR) and platelet-derived growth factor receptor (PDGFR), two well-known markers for glioblastoma aggressiveness. Therefore, we studied whether the expression of key actors of macropinocytosis is modified in human glioma datasets. Strong deregulation has been evidenced at the mRNA level according to the grade of the tumor, and 38 macropinocytosis-related gene signatures allowed discrimination of the glioblastoma (GBM) samples. Honokiol-induced vacuolization was then compared to vacquinol-1 and MOMIPP, two known macropinocytosis inducers. Despite high phase-contrast morphological similarities, honokiol-induced vacuoles appeared to originate from both endocytosis and ER. Also, acridine orange staining suggested differences in the macropinosomes’ fate: their fusion with lysosomes appeared very limited in 3-(5-methoxy -2-methyl-1H-indol-3-yl)-1-(4-pyridinyl)-2-propen-1-one (MOMIPP)-treated cells. Nevertheless, each of the compounds markedly increased temozolomide uptake by glioma cells, as evidenced by LC-MS. In conclusion, the observed deregulation of macropinocytosis in GBM makes them prone to respond to various compounds affecting their formation and/or intracellular fate. Considering that sustained macropinocytosis may also trigger cell death of both sensitive and resistant GBM cells, we propose to envisage macropinocytosis inducers in combination approaches to obtain dual benefits: increased drug uptake and additive/synergistic effects.

## 1. Introduction

Glioblastoma (GBM) remains associated with a dismal prognosis, partly due to a high level of resistance to proapoptotic stimuli and a propensity for cell migration and invasion of normal brain tissue [1,2]. Although the majority of chemotherapeutic drugs operate by triggering caspase-dependent apoptotic cell death, 17 cell death types have been described to date [3,4,5], including autophagy. The latter is a programmed pathway that can either promote or hinder tumor development. It is triggered by several anti-tumor agents, including temozolomide (TMZ), particularly in cancer cells lacking essential apoptotic modulators, such as GBM [6]. The autophagosomes arise under the control of the PI3K-Akt pathway—probably from diverse sources, including endoplasmic reticulum (ER), Golgi and plasma membrane—and further fuse with lysosomes to digest their contents [5]. To date, autophagy remains the best way to combat GBM, since it is induced by the standard-of-care drug TMZ [7]. Other forms of non-apoptotic cell death include methuosis, paraptosis, necroptosis, oncosis and lysosomal membrane permeabilization plus senescence. All these forms of cell death are associated with accumulation of cytoplasmic vacuoles (vacuolization) [3,5]. Irreversible vacuolization marks cytopathological conditions leading to cell death; however, the size, content, function and origin of the vacuoles may differ markedly from one type of cell death to another.

One mechanism that can lead to cellular vacuolization is dysregulation of macropinocytosis. Macropinocytosis is a clathrin-independent endocytic mechanism that was first described by Warren Lewis in 1931 [8]. This process is characterized by the nonselective uptake of extracellular fluid [9]. Macropinosomes consist of large vesicles (0.2 µm < diameter < 5 µm), formed by actin-mediated ruffling of the plasma membrane. The nascent macropinosomes migrate in a centripetal manner and rapidly acquire markers of late endosomes, such as Rab7, before their fusion with the lysosomal compartment [9]. Cancer cells generally employ macropinocytosis to internalize cell surface receptors and certain nutrients [10]. Starving tumors by inhibiting macropinocytosis was recently suggested as a potential therapeutic strategy to combat Ras-driven cancers [11]. Deregulation of macropinocytosis in cancer cells seems to be correlated with oncogenic *RAS* [12], and more specifically *K-RAS* in clinical samples [13,14], even if induction of Src may also trigger macropinocytosis [15,16]. With respect to glioma, high levels of active wild-type Ras have been reported in high grade astrocytomas, but oncogenic mutations in *H-RAS*, *K-RAS* or *N-RAS* are rare (occurring in less than 2% of the cases) [17]. Another major pathway known to stimulate macropinosome formation is under the control of the epidermal growth factor receptor (EGFR) and platelet-derived growth factor receptor (PDGFR) [18]. Amplification of wild-type or mutant *EGFR* is relatively common (43% of GBM cases), with the most frequent mutation being *EGFRvIII*. Mutation and/or overexpression of *EGFR* or *PDGFR* are very well known markers of glioma malignancy [19,20]. Importantly, EGF-mediated macropinocytosis appeared dependent on c-Src activity [16].

Despite the potential importance of macropinocytosis for survival of glioma cells, little information is currently available about the possible deregulation of macropinocytosis in these tumors. The possible relevance of macropinocytosis in gliomas is underscored when considering that it contributes, at least partly, to the uptake of extracellular vesicles (particularly exosomes) produced by glioma cells to enhance cancer progression and promote angiogenesis, metastasis, immunosuppression and chemoresistance [21,22]. While active macropinocytosis may represent a survival strategy for gliomas, it may also be exploited to increase the intracellular delivery of therapeutics (soluble as well as vesicular-formulated ones) [22,23,24,25]. Moreover, there is strong evidence that hyperstimulation of macropinocytosis by Ras overexpression or by treatment with small molecules can be detrimental to glioma cells, leading to a form of cell death termed as methuosis [26]. Normally, macropinosomes are formed from plasma membrane ruffling via Rac1-dependent rapid polymerization/depolymerisation of branching actin filaments. The macropinosomes then undergo a maturation process wherein they are either degraded via a late endosome/lysosome process or recycled back to the plasma membrane [23]. During abnormal macropinocytosis, as observed in methuosis, macropinosomes do not fuse with lysosomes or recycle to the plasma membrane. Instead, they fuse with each other to form multiple larger vacuoles that exhibit properties of non-functional late endosomes [27]. Through mechanisms that are yet to be elucidated, the extreme vacuolization ultimately leads to death and rupture of the cell. While Ras-induced methuosis is accompanied by caspase activation, cell death cannot be prevented by caspase inhibitors. Methuosis is therefore considered to be caspase-independent [28]. Although early studies demonstrated that methuosis can be induced in glioma by overexpressing active forms of Ras or Rac1, more recent studies have identified small molecules that can trigger this form of death in a manner that is independent from Ras or Rac1. Specifically, the indolyl chalcone 3-(5-methoxy -2-methyl-1H-indol-3-yl)-1-(4-pyridinyl)-2-propen-1-one (MOMIPP), which was used in the present study, can induce intense macropinocytosis, leading to methuosis in cultured GBM cells at low micromolar concentrations [28]. Similar results were obtained in TMZ-resistant glioma cells [29].

Thus, this study aims to (i) evaluate in a systematic manner whether genes implicated in macropinocytosis are deregulated in gliomas, employing analyses of mRNA expression levels in clinical sample datasets and (ii) evaluate whether the induction of severe vacuolization and macropinocytosis with small molecules may represent an interesting strategy to enhance the uptake and/or efficiency of chemotherapeutic agents by gliomas as detailed in the Discussion section.

## 2. Results

### 2.1. Evaluation of The Deregulation of Macropinocytosis in Human Gliomas

To evaluate whether macropinocytosis is deregulated in gliomas, we made use of mRNA expression databases of human samples. Table A1 (Appendix B) summarizes the 38 selected main proteins participating in either the initiation/regulation or the formation and recycling of macropinosomes, with their detailed function in that process. The GDS4290 Henry Ford database regroups the mRNA expression array data of 180 patients: 23 non-tumoral samples (from epilepsy patients) and 157 glioma samples. We first performed a systematic, statistical comparison of the mRNA expression of the 38 targets between normal brain samples (*n* = 23), grade II (*n* = 45), grade III (*n* = 31) and grade IV (*n* = 81) glioma samples through non-parametric Kruskall–Wallis comparison and two-tailed tests. Detailed results per probe, provided in Table A1, highlighted that 14/38 targets were overexpressed in glioma while 9/38 were downregulated. This means that more than 60% (23/38) of genes playing key roles in macropinocytosis are deregulated in glial tumors.

As expected, *EGFR* and *PDGFR* were upregulated according to the malignancy grade of the tumor (Figure 1a,b), but other key players in macropinocytosis and methuosis were also upregulated according to the grade of the tumor, including actin dynamic-related gene products (*SWAP70*, several *ARPC* involved in the Arp2/3 complex), their upstream activator adenosine diphosphate ribosylation factors 6 (*ARF6*), and actin-rich membrane ruffles associated protein Rab34 (Figure 1d,f,h). In contrast, the expression level of GIT1 that mediates the deactivation of Arf6 is decreased (Figure 1c), along with the other genes involved in the closure of the macropinosomes, such as *PAK1*, *CTBP1* and *PDL1* (Table A1, Figure 2). *RAB20* and *SNX5*, two markers of initial macropinosome maturation, are both upregulated (Table A1; Figure 1e,g; Figure 2). Thus, the expression of genes encoding numerous proteins associated with macropinocytosis is deregulated in glioma in comparison to normal brain tissues; these changes are additionally correlated with the grade of the tumor (Figure 1). Figure 2 schematically illustrates the macropinocytosis process, noting the proteins analyzed and the changes observed in their expression in glioma tumors. According to the high number of targets whose mRNA levels were found deregulated, we proceeded with an unsupervised hierarchical clustering of the 180 patient samples on the basis of those 38 gene expression signatures, and we observed a clear tendency of grouping according to the grade of the sample (Figure 3a). While normal brain tissue samples grouped on the left, the GBM samples were distinctly clustered on the right of the Euclidian-based tree (Figure 3a). Note that while grade II samples also appeared grouped between non-tumoral brain samples and grade IV samples, grade III samples were spread among grade II or grade IV samples (Figure 3a).

In order to evaluate whether this deregulation has any impact on the survival of the patient, we carried out the same analysis on the data from the GDS53733 database of 70 samples of grade IV GBM patients (16 short term survival patients (<12 months); 23 long term survival patients (>36 months); and 31 intermediate survival patients). We found only very limited statistically significant changes, i.e., with respect to probes for *SWAP70* and *CYFIP1* and, to a lesser extent, *PDGFRA*, *EGFR* and *ABI1* (Appendix A). Nevertheless, the unsupervised heatmap clustering revealed a tendency of grouping of the samples from long surviving GBM patients into a subcluster (Figure 3b). In addition, we noticed that many targets overexpressed in GBM samples of the 180 series were downregulated in long surviving patients, including *RAB20*, *RAB34*, *SWAP70*, *CYFIP1*, *ARPC1B*, *ARPC2* and *ARPC4*. The dataset is certainly limited in size and further investigation integrating all clinical aspects should be conducted to determine whether deregulated macropinocytosis could be a predictive marker for clinical outcome of the patient.

### 2.2. Morphological Comparison of Glioma Cells Treated by Honokiol, 3-(5-methoxy -2-methyl-1H-indol-3-yl)-1-(4-pyridinyl)-2-propen-1-one (MOMIPP) and Vacquinol-1

Next, we examined the effects of three compounds that alter vesicle trafficking or autophagy in glioma cells. Two are known for their propensity to induce macropinocytosis—i.e., vacquinol-1 [27] and MOMIPP [29]—while the last one, honokiol, is a natural bioactive polyphenol extracted from several parts of Magnolia genus tree [30] (Figure 4). The latter displays various well-known pharmacological properties, such as anti-oxidant [31], neuro-protective [32] and anti-inflammatory effects in both microglia and astrocytes [33]. These properties explain the traditional use of honokiol for the treatment of thrombotic stroke, gastrointestinal complaints, anxiety and nervous disturbance [34]. Honokiol has also been shown to display interesting anti-tumoral properties against glioma cells, where it may trigger p53-mediated cell cycle arrest and apoptosis [35] or alternative autophagic cell death [36].

Surprisingly, we observed striking cytoplasmic vacuolization in cells exposed to honokiol at concentrations slightly lower than its IC_50_ in Hs683 (40 µM) and U373 (35 µM) glioma cells (Figure 5 with respect to the U373 cell line; Appendix A for Hs683 cells). Although the vacuolization appeared delayed with honokiol in comparison to MOMIPP and vacquinol-1, the progressive increase in the number and size of highly refringent vacuoles and their apparent homotypic fusion resembled the morphological features observed with the macropinocytosis inducers at equipotent concentrations (close to their own IC_50_ as determined by means of MTT colorimetric assay, i.e., 3 µM for MOMIPP and 5 µM for vacquinol-1; see Appendix A for the MTT curves).

### 2.3. Characterization of the Vacuoles Induced by Honokiol, Vacquinol-1 and MOMIPP

To obtain more insight regarding these vacuolization processes, we utilized different inhibitors of vacuolization, i.e., bafilomycin A1, ethyl-isopropyl-amiloride (EIPA), EHT1864, PP2 and filipin.

Bafilomycin A1 is an inhibitor of the vacuolar H^+^-ATPase that plays crucial roles in maintaining low pHs of late endosomes and lysosomes. Bafilomycin A1 was accordingly shown to block the endosomal and endosome-lysosomal fusion during macropinocytosis [37]. Additionally, bafilomycin A1 has been suggested to also inhibit nascent macropinosome formation, similarly to the Na^+^/H^+^ exchanger inhibitor, by disrupting the fine tuning of submembranous pH needed for the recruitment and activation of Rac1 and Cdc42 to membrane ruffles [38]. As illustrated in Figure 6 with respect to the U373 cell line (Appendix A for HS683 cells), bafilomycin A1 almost completely inhibited the vacuolization induced by each of the three compounds. Consistently, similar results were obtained with respect to the Na^+^/H^+^ exchanger inhibitor EIPA [9] when its concentration adapted to the cell line and the compound under investigation in accordance with the timing of the vacuolization that they induced (Figure 6 and Appendix A).

In contrast, the Rac1 inhibitor, EHT1864 [26], has only slight effects on honokiol and vacquinol-1-induced vacuolizations and no effect on MOMIPP–induced vacuolization (Figure 6 and Appendix A). Alternatively to oncogenic RAS stimulation, Src was also demonstrated to participate in macropinosome membrane ruffling via PI3K [15,16]. We therefore made use of the Src inhibitor PP2 [39]. This later inhibited vacuolization induced by honokiol at 25 µM (Appendix A). We tried higher concentrations to possibly inhibit MOMIPP and vacquinol-1 vacuoles but encountered solubility issues making us impossible to conclude at this stage whether Src may or not be involved in their vacuolization processes.

The last inhibitor, filipin, is a cholesterol binding agent [26] known to inhibit clathrin-independent endocytosis. Consistently, macropinocytosis occurs in cholesterol-rich membrane domains [40]. When used at nontoxic concentrations, i.e., 1 µg/mL, filipin had no effect on the vacuole formation induced by each of the three compounds (Appendix A). Previous studies demonstrated that filipin effectively impaired MOMIPP-induced vacuolization in a different cell line, but it was at higher concentrations, i.e., 12 µg/mL. However, in our cellular models, such high concentrations proved to be too toxic.

Finally, we utilized fluorescent probes to further decipher the origin of vacuoles. Lucifer yellow is a fluid-phase tracer that is internalized intracellularly by endocytic processes including macropinocytosis [12]. We observed, as expected, that both MOMIPP and vacquinol-1 increased the number of positive lucifer-yellow vacuoles—a feature that was also observed with honokiol, but to a lesser extent, with several vacuoles remaining negative (Figure 7a and Appendix A).

As macropinocytosis differs from other endocytic processes by its capacity to internalize large extracellular volumes and high molecular weight molecules, we evaluated this feature by means of 10 kDa and 70 kDa fluorescent dextran staining. Figure 7b and Appendix A show that each of the three compounds seemed to trigger the uptake of 10 kDa dextrans that was observed in small vacuoles, but rarely in the most enlarged ones. MOMIPP and vacquinol-1 seemed to uptake 70 kDa dextrans, contrary to honokiol, which had no positive vacuole (Appendix A). A quantitative assay was conducted with 10 kDa Texas Red-labeled dextran. Significant increased uptake under treatment with each of the three compounds was confirmed, except in the case of U373 cells treated 24 h with honokiol (Figure 7c), and the most marked increase was obtained with vacquinol-1. By contrast, acridine orange labelled most of the enlarged vacuoles induced by honokiol in red, suggesting acidic content (Figure 8 and Appendix A). Interestingly, while only few of MOMIPP-induced vacuoles were also stained red (Figure 8 and Appendix A), vacquinol-1-induced vacuoles were positive for both red and green fluorescence after acridine orange staining [41]. When looking for lysosomal staining, these organelles were readily observed in cells treated either with honokiol or MOMIPP (Appendix A). Thus, the principal difference between these was that most of the large vacuoles remained unstained by acridine orange with MOMIPP, whereas most of them appeared acidic in honokiol-treated cells while being lyso-Tracker negative (acidic; Appendix A). This could suggest that the large vacuoles induced by honokiol might result from fusion of endosomes with lysosomes or autophagolysosomes at some point during their biogenesis, while MOMIPP completely blocks vacuole fusion with lysosomes (Figure 8 and Appendix A). Finally, although vacquinol-1 appears to trigger the strongest increase in macropinocytosis tracers (Lucifer Yellow and high molecular weight dextrans), few enlarged vacuoles have been observed, suggesting different effects on macropinosome maturation and recycling.

We also used probes for mitochondria and ER to assess the contributions made by these compartments to the vacuoles induced by honokiol. We observed some ER-positive vacuoles only in the honokiol-treated cells (Figure 8 and Appendix A). Both mitochondria and ER are thought to be a source of vacuolization during paraptosis [42]. Thus, honokiol differed from both MOMIPP and vacquinol-1 and could induce paraptosis features in these glioma cell lines, similar to what was previously described in leukemia cells [43]. ER swelling does not play any role in macropinocytosis. Accordingly, ER-positive vacuoles were still present after treatment with the inhibitors of macropinocytosis (Bafilomycin A1, EIPA and PP2; Appendix A). Note that none of the treatments triggered mitochondrial swelling (Appendix A). This supports the hypothesis that honokiol may induce paraptosis in addition to macropinocytosis according to the other organelle markers detailed above (see Discussion section).

### 2.4. Evaluation of the Effects of Honokiol, Vacquinol-1 and MOMIPP on Intracellular Temozolomide (TMZ) Concentration

To highlight the possibility of using these inducers of vacuolization to enhance the penetration of chemotherapeutic agents in cancer cells, we pre-treated glioma cells with the three compounds of interest before the addition of the treatment with TMZ. TMZ was chosen as the current first-line chemotherapeutic agent against newly diagnosed GBM [44,45] and for its pro-autophagic effects in glioma cells [46,47]. Honokiol was used 22 h, vacquinol-1 15 h and MOMIPP 3 h before the addition of TMZ. These timepoints were selected according to the time required to visualize the beginning of the vacuolization with each compound. The quantification of intracellular TMZ in Hs683 and U373 cells was achieved through HPLC-MS after two hours of subsequent exposure to TMZ at 200 µM with or without the macropinocytosis inducers. Figure 9 shows that all three compounds significantly increased the intracellular TMZ concentration in both Hs683 and U373 cells, except in the case of MOMIPP-pretreated Hs683 cells. Note that we pre-treated cells only for 3 h with MOMIPP, and longer incubation periods may be required in these cells.

Despite apparent differences in their effect, certain considerations, including the concentration of the compound itself, the duration of the pretreatment required, or their ability to cross the blood brain barrier (BBB), are all crucial to be taken into account for future in vitro and in vivo combinations. Furthermore, this experiment only highlighted that the use of those compounds could help to increase the intracellular concentration of TMZ. Whether these effects actually related to macropinocytosis induction and/or other mechanisms should be evaluated in more depth. Such investigations would help decipher whether this strategy could be used with other kinds of drugs and pharmaceuticals. Other aspects and advantages of combinations (pro-methuotic or pro-paraptotic effects, additive or synergisctic effects) should also be considered, as discussed below.

## 3. Discussion

In this study, we showed for the first time that genes associated with macropinocytosis were deregulated in human glioma brain tumors. This conclusion is based on a systematic statistical analysis of the mRNA dataset from Henry Ford Hospital by regrouping non-tumoral human samples in comparison to gliomas samples of grades II, III and IV. Figure 2 summarizes the genes of the macropinocytosis process whose expression levels were evaluated in this study and whether they were up- or downregulated. Hereafter, we discuss these results briefly in relation to the scientific knowledge currently available regarding the main deregulated targets in glioma biology.

Consistent with existing literature, we found *EGFR* and *PDGFR* to be upregulated in gliomas [19,20,48]. These are potent inducers of macropinocytosis [18]. *H-RAS*, by contrast, was found to be downregulated, but only the mRNA level was investigated in the present study. Indeed, increased active wild-type Ras activity has been reported in high-grade glioma and may contribute to increased macropinocytosis as well [17]. Interestingly, Src plays key roles in EGF-triggered macropinocytosis associated with enhanced migration and further fusion of macropinosomes with lysosomes [16]. In this study we observed that the Src inhibitor PP2 impaired acid vacuolization induced by honokiol (see below).

We also found that several other key actors of macropinocytosis, such as Rab34, Arp2/3 complex and SWAP70, were significantly overexpressed according to the grade of the tumor (Figure 1, Table A1). These are all implicated in the organization of the actin cytoskeleton and ruffling of the plasma membrane during macropinocytosis [49,50,51]. Our results are consistent with previous data suggesting that overexpression of these proteins contributes to glioma cell invasion [52]. In this context, expression levels of *ARP2/3* and *RAB34* correlate with the grade of the tumor and/or the survival of the patient, with both being upregulated from low-grade to high-grade gliomas [53,54]. *RAB34* was additionally associated with poor patient survival [54]. Regarding other Rab proteins, only few studies have been published regarding their roles and/or deregulation in glial tumors. The increased expression of *RAB20* found in the present study could suggest a role in macropinocytosis in glioblastoma. Downregulation of *RAB21* by siRNA significantly inhibited cell proliferation and remarkably induced cell apoptosis [55], but its overexpression in glioblastoma was not reported, and no link to enhanced macropinocytosis was suggested in those tumors. In addition, mRNA coding for Arf6 was also found to be overexpressed in GBM samples (Figure 1d). During macropinocytosis, active Arf6 recruits the Arf nucleotide binding site opener (ARNO) that activates Arf1 via its guanine exchanger activity and allows the recruitment of the WAVE protein regulatory complex (WRC). This heteropentameric complex of WASP family proteins composed of WAVE, Cyfip1, Nap1 (*NCKAP1*), Abi1 and Brk1, in turn activates Arp2/3 to initiate actin polymerization [51,56]. Recently, a potential role for Arf1 in glioblastoma progression was suggested [48], and over-expression of Arf6 was shown to enhance glioma cell migration both in vitro and in vivo [48,57,58].

While we did not observe significant changes in *LAMP1* expression, Jensen et al. [59] found that *LAMP1* is more highly expressed in glioblastoma than in diffuse and anaplastic astrocytomas, even though its expression does not correlate with the overall survival of the patient [59,60]. *CTBP1/BARS* and its downstream target *PLD1*, whose activation triggers the closure and the final fission of macropinosomes from the plasma membrane [9,61,62], were found downregulated in high-grade tumors at the mRNA level herein. Those results are not consistent with previous data indicating positive correlation of Ctbp protein antigen expression with the histopathologic grade of the glioma [63] and worse survival of the patient [64]. The discrepancies could be related to evaluation at the mRNA versus protein expression level. Further, the downregulation of *PAK1* mRNA is not easy to integrate when considering that both protein phosphorylation and localization (cytoplasmic versus nuclear) appeared essential for its function and prognosis value in glioblastoma [65].

Nevertheless, as highlighted by Figure 2, the macropinocytosis process appeared obviously deregulated in glioma, and particularly in GBM (Figure 1). Accordingly, we showed that the mRNA expression signature of these 38 genes taken together, and covering most of the macropinosome formation, maturation and turn-over processes, enabled discriminating GBM from non-tumoral samples and lower grade glioma on the basis of unsupervised analysis (Figure 3). Macropinocytosis could, thus, participate in GBM aggressiveness, notably when considering its contribution for nutrient uptake and exosome GBM crosstalk [21,22]. Even if a systematic comparison of each target alone, according to the survival of GBM patient, did not reveal any significant changes (Appendix A), long surviving GBM patients displayed a trend of grouping together on the basis of these 38 genes’ signatures. A study at the protein level, and with full patient characterizations and follow-ups, should be conducted to further confirm an upregulation of macropinocytosis in GBM, and whether it might be of prognosis value for the patient.

Recent findings have suggested that stimulation of macropinocytosis in cancer cells can lead to increased chemotherapeutic efficiency [22,23,24,25,66]. Thus, we herein propose taking advantage of this possible Achilles’ heel by the use of macropinocytosis-stimulating agents to combat GBM. Hyperstimulation of macropinocytosis with small molecules may ultimately induce methuotic cell death in glioma cells, as previously shown [26]. However, when used for short periods of time or at sub-lethal concentrations, methuosis-inducing compounds may be useful for increasing the intracellular concentrations of chemotherapeutic agents.

Therefore, we compared the effects of three potential compounds of interest—honokiol, vacquinol-1 and MOMIPP. We observed high morphological similarities between the vacuolization processes induced by those three compounds in terms of refringency, increase in number followed by apparent fusion, and cytoplasmic accumulation (Figure 5 and Appendix A). Although they are all inhibited by both bafilomycin A1 and EIPA, one unique inhibitor of macropinocytosis, our results indicate that the vacuoles they induce may differ in terms of origin and capacity to fuse with lysosomes. Vaquinol-1 actually appeared to stimulate macropinosome formation according to the marked increase in the uptake of Lucifer Yellow and dextrans (Figure 7 and Appendix A). The fact that it induces vacuoles of intermediate pH, as revealed by acridine orange staining (positive in both red and green fluorescences; Figure 8 and Appendix A) and the absence of increase in lysosomal content, are two features that differentiate vacquinol-1 from MOMIPP- and honokiol-induced effects. Indeed, these two latter display slight increases in lysosomal content assessed by Lyso-tracker. However, the enlarged vacuoles are not stained with that tracker. (Appendix A). Such enlarged vacuoles were not obtained with vacquinol-1 (Figure 5 and Appendix A). Important differences were also obtained with respect to uptake and acidity evaluated by acridine orange staining. In the case of honokiol, most of those large vacuoles appeared acidic as revealed by their red staining with acridine orange, while MOMIPP-induced enlarged vacuoles remained negative. We conclude that the three compounds affect macropinocytosis process and/or endosomal traffic in high-grade glioma cells, but that the mechanisms underlying those effects markedly differ from one compound to the other. In particular, the fusion of the vacuoles with lysosomal compartments seems to still occur with respect to honokiol, and partly to vacquinol-1 treatments, but not with MOMIPP. In addition, honokiol is the only compound in the current study that also induces ER-derived vacuolization (Figure 6 and Appendix A). Note that ER stress has been shown previously to inhibit endocytic pathways, including macropinocytosis [67,68]. This effect may also be linked to the ability of honokiol to induce paraptosis-like cell death, as previously reported in leukemia cells [43], and autophagy in glioblastoma [36,69]. Interestingly, vacquinol-1 was also recently shown to trigger mitophagy in GBM cells [70]. Although both macropinocytosis and autophagy represent two “opposite” major mechanisms to provide nutrient supply and recycling from extracellular versus intracellular sources respectively, they seem to share upstream and downstream regulation whose links remain to be fully deciphered [71]. In contrast, methuosis results from the accumulation of unprocessed macropinosomes that fuse together, rather than fusing with lysosomes or recycling to the plasma membrane [27], as described previously with respect to the MOMIPP chalcone.

Obviously, further studies are required to clarify if and how each of these compounds affect membrane ruffling, macropinosome formation, trafficking, fusion, and recycling independently to other effects such as ER swelling or autophagy.

Nevertheless, we observed that all three compounds were able to significantly increase intracellular TMZ concentration, even if TMZ uptake was excellent [47]. This study provides a proof of concept, and it encourages further investigations with drugs whose intracellular and/or BBB penetrations are more problematic. This could be the case with respect to antibody-based therapies, including depatuxizumab mafodotin, a new antibody-drug conjugate with promising clinical results notably in EGFR-amplified GBM cases [72,73]. Typically, patients with EGFR amplification may benefit from macropinocytosis-interfering drugs, such as those studied herein.

Honokiol and MOMIPP have been proven to cross the BBB [35,74,75]. Although vacquinol-1 also displays an adequate BBB penetration, its systemic toxicity requires a lowering dose or local delivery [76]. At a tolerable dose, vacquinol-1 alone allowed a reduction in tumor size, but it did not increase the overall survival of the GBM preclinical model [76]. Thus, toxicological issues have to be taken into account for future development. Honokiol is widely used and freely available as a phytotherapeutic complement in several countries for medicinal properties against anxiety, for facilitation of sleep, support of cognitive functions, and antioxidant effects. Further, possible additional and/or synergistic effects between macropinocytosis inducers and chemotherapeutic agents may be expected. Accordingly, honokiol has already been shown to increase TMZ cytotoxic effects in vitro [77]. Both honokiol and MOMIPP have been shown to kill drug-sensitive and -resistant glioma cells [29,77]. Recent studies pointed JNK Kinase as an important signal for both honokiol-induced effects on stem cells [78] and cytotoxic effects of indoylchalcones such as MOMIPP [75].

Numerous perspectives remain to be addressed, including the following: (1) evaluation of whether macropinocytosis deregulation may be linked to patient survival and response to chemotherapy, (2) deciphering how each compound affects macropinocytosis and other endocytosis processes, and (3) testing the efficiency of combined treatments for the proposed molecules investigated herein with chemotherapeutic agents, with a particular attention to new therapeutic drugs characterized by limited intracerebral brain pharmacokinetics.

## 4. Materials and Methods

### 4.1. Cell Lines and Compounds

The Hs683 human oligodendroglioma cell line was obtained from the American Type Culture Collection (ATCC, code HTB-138) and the human glioblastoma U373 cell line from the European Collection of Cell Culture (ECACC, code 08061901). Cells were cultivated in RPMI 1640 (Gibco, Thermofisher, Dilbeek, Belgium) culture medium supplemented with 10% heat-inactivated fetal bovine serum (Gibco), 0.6 mg/mL L-glutamine (Gibco), 200 IU/mL penicillin–streptomycin (Gibco), and 0.1 mg/mL gentamicin (Gibco) at 37 °C with 5% CO_2_. Cultures were checked twice a month for mycoplasma.

Honokiol was purchased from Sigma-Aldrich (St. Louis, MO, USA) as well as EIPA and PP2, Vaquinol-1 and EHT1864 from Selleckchem (Houston, TX, USA), Bafilomycin A1 from Cayman chemicals (Selleckchem), and filipin complex from AG Scientific (San Diego, CA, USA). MOMIPP was synthesized and characterized as previously described [29].

### 4.2. MTT Colorimetric Assay

Cell viability was determined using a colorimetric assay as described previously [79]. Cells were seeded in 96 well plates (Sarstedt AG & CO, Nümbrecht, Germany) and were grown for 24 h. They were then treated with honokiol, vacquinol-1, or MOMIPP at concentrations ranging from 0.01 to 100 µM or left untreated for 72 h. The viability was estimated by using 3-(4,5-dimethylthiazol-2-yl)-2,5-diphenyl tetrazolium bromide (MTT, Sigma-Aldrich) mitochondrial reduction into formazan measured at 570 nm with a spectrophotometer (680XR, Bio-Rad Laboratories, Berkeley, CA, USA; reference wavelength 610 nm). The experiment was realized two times in sextuplicate for each cell line and each compound.

### 4.3. Characterization of Vacuoles

#### 4.3.1. Phase Contrast Microscopy for Morphological Observations

In order to observe the morphological changes induced by treatments, pictures of living cells were taken with the Imager M2 fluorescence microscope (Carl Zeiss, Zaventem, Belgium) coupled with the AxioCam ICm1 and AxioImager software (Carl Zeiss). Cells were seeded on glass coverslips in six-well plates (Sarstedt AG & CO) and allowed to attach and start growing for 24 h. Afterwards, cells were either left untreated or treated with each compound as follows: 35 µM and 40 µM of honokiol for U373 and Hs683 cell lines, respectively; and 5 µM of vacquinol-1 and 3 µM of MOMIPP for both cell lines.

For the characterization of the vacuolization by using inhibitors, cells were either pre-treated 1 h with bafilomycin A1 (100 nM), filipin (1 µg/mL), or co-treated with EHT1864 (25 µM), EIPA (10, 30 or 75 µM) or PP2 (25 µM). At different time points, coverslips were washed twice in PBS, transferred onto microscope slides, and four pictures were taken per slide. Each experimental condition was tested twice in triplicate.

#### 4.3.2. Fluorescent Microscopy Assays

Fluorescent probes that stained different cellular compartments were used to characterize the origin of the vacuoles. The dapoxyl ER-tracker blue-white dye, the Lyso-tracker red dye, the Mito-tracker green dye, and 10 kDa and 70 kDa Texas-Red labeled dextran were all obtained from Molecular Probes (Life Technologies, Merelbeke, Belgium). We also used Lucifer yellow (Lucifer Yellow CH, lithium salt) from Biotium (Fremont, CA, USA) and acridine orange from Sigma-Aldrich. Briefly, the cell seeding and treatment procedures were similar to the ones used for the phase contrast microscopy (Section 4.3.1.). The dye solutions were simply added to the culture medium 1 h before the end of the treatment periods, excepted for the Lucifer yellow and both dextrans 10 kDa and 70 kDa, which were added for the whole duration of the treatment. The concentrations of the dyes were as follows: ER tracker, 0.5 µM; Lyso tracker, 75 nM; Mito tracker, 200 µM; Lucifer yellow, 100 µg/mL; acridine orange, 1 µg/mL; and dextrans 10 kDa and 70 kDa, 125 µg/mL; At the end of the treatment period, the procedure was similar to that of phase-contrast microscopy to take pictures of living cells with the Imager M2 fluorescence microscope (Carl Zeiss) coupled with the AxioCam ICm1 and AxioImager software (Carl Zeiss). The experiment was realized at least two times in duplicate.

#### 4.3.3. Flow Cytometry

In order to quantify the internalization of Texas-red labeled dextran 10 kDa (Molecular Probes), cells were treated with honokiol (35 µM for U373 and 40 µM for Hs683 cells), MOMIPP (3 µM), or vacquinol-1 (5 µM), or left untreated in the presence of Texas-red labeled dextran 10 kDa (125 µg/mL). After 24 h for U373 and 30 h for Hs683, cells were washed twice with PBS, detached with trypsin-EDTA (Gibco) and centrifuged. The supernatant was removed and cells were resuspended in 250 µL of PBS for flow cytometry analysis with the Beckmann Gallios apparatus (Beckmann Coulter, Analis, Suarlee, Belgium). Each sample recorded 10,000 events, and the experiment was conducted once in sextuplicate.

### 4.4. Quantification of Intracellular TMZ

#### 4.4.1. Sample Preparation

For the experiment, cells were seeded in T75 flasks (Sarstedt AG & CO). When the confluence of the flasks was around 75%, cells were pre-treated with honokiol (35 µM for U373 and 40 µM for Hs683), MOMIPP (3 µM), vacquinol-1 (5 µM), or left untreated for different periods of time (3, 15 and 22 h) before the addition of TMZ (200 µM) for 2 h. After the treatment, the culture medium was removed, the cells were washed twice with cold PBS (Gibco)), scrapped in 200 µL of ice-cold methanol (VWR International, Oud-Heverlee, Belgium), and sonicated for 30 s. As TMZ is stable at pH < 4 [80], 50 µL of cell lysate was diluted in 50 µL of an acid internal standard solution (2 µM theophyline and 0.1% formic acid in methanol).

#### 4.4.2. LC-MS Process, Data Acquisition and Analysis

Ten microliters from the prepared sample were injected into the liquid chromatography (LC) system. Analyses were performed with a rapid resolution LC (RRLC) 1200 series from Agilent Technologies (Santa Clara, CA, USA). Separation was performed on a Zorbax Eclipse XDB-C18 Rapid Resolution HT column (4.6 × 50 mm, 1.8 µm particle size) from Agilent Technologies, preceded by a Zorbax Eclipse XDB-C18 pre-column (4.6 × 5 mm, 1.8 µm particle size) using water supplemented with a 0.1% formic acid/acetonitrile gradient. A 6520 series electrospray ion source (ESI)–quadrupole time-of-flight (QTOF) from Agilent Technologies was used for the MS analyses. Initial ESI–Q-TOF parameters were as follows: positive mode; capillary voltage of 4500 V; dynamic high resolution acquisition mode (2 Hz); gas temperature of 350 °C; drying gas flow of 9 L/min; nebulizer pressure of 50 psig; fragmentor voltage of 130; and skimmer voltage of 65 V. Data was acquired using the Mass Hunter Acquisition software (Agilent Technologies, version B.04 SP3) and analyzed by Mass Hunter Quantitative Analysis software (Agilent Technologies, version B.07 SP1). TMZ was monitored at an m/z value of 195.0625 and theophyline at an m/z value of 181.0720. A quantitative curve was performed over the range 0.1 to 10 µM of TMZ.

### 4.5. Statistical Analyses

The mRNA expression analyses were conducted on two datasets of human samples publicly available on the NCBI GEO repository. The GSE4290 dataset, from the Henry Ford Hospital, was published in 2006 [81]. The GSE53733 dataset regroups the mRNA expression data sets of 70 human glioblastoma (grade IV) primary tumors from the German Glioma Network and was published in 2014 [73]. Both expression data were generated using the same microarray platform (Affymetrix Human Genome U133 Plus 2.0 Array).

The microarray analysis was based on the robust multi-array average (RMA) expression values, which were obtained with affy package v1.56 of Bioconductor/R packages (http://www.bioconductor.org). The heatmaps were made using the pheatmap R package v1.0.10. The samples were grouped by annotation or clustered using hierarchical clustering using the average linkage on the Euclidian distance.

In both databases, we collected the data corresponding to the probes of 38 target genes involved in macropinocytosis (see list in Table A1). Comparison of the expression level for each probe was conducted using the Kruskal–Wallis test (a nonparametric, one-way analysis of variance), followed by two-tailed tests using Statistica (Statsoft, Tulsa, OK, USA).

For in vitro biological assay comparisons (flow cytometry and TMZ quantification), we conducted non-parametric Mann–Whitney tests using Statistica.

## 5. Conclusions

In conclusion, we found that more than 60% of the 38 macropinocytosis-related genes studied herein are overexpressed or down-regulated in GBM patient samples. These 38 genes may constitute a signature discriminating GBM from non-tumoral samples and lower grade gliomas. Those results suggest that macropinocytosis may play important roles in GBM aggressiveness.

We then proposed to make use of compounds that interfere with this endocytotic process to increase anti-cancer drug uptake. The three compounds selected for this purpose, i.e., honokiol, vacquinol-1 and MOMIPP indeed allowed significant increase in intracellular TMZ concentration in vitro. This study provides with a first proof of concept and paves the way to use macropinocytosis deregulators in combination with chemotherapeutic agents. Considering that EGFR is an activator of macropinocytosis, patients with amplified or mutated EGFR may represent better candidate for this approach.

## Figures and Tables

**Figure 1 cancers-11-00411-f001:**
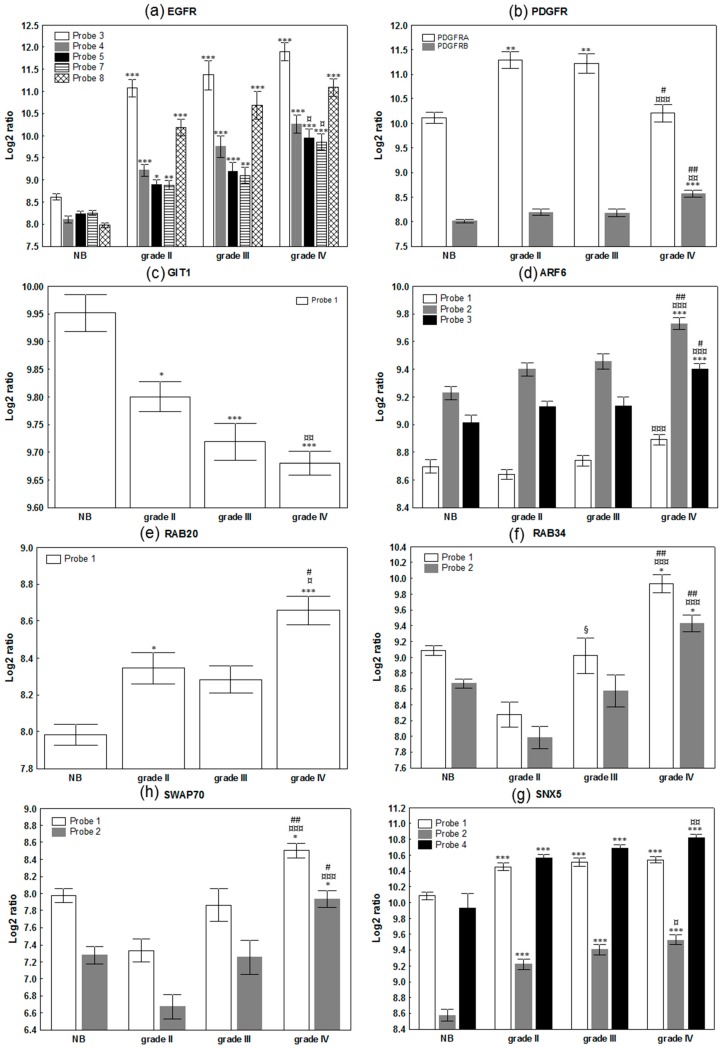
mRNA expression level of selected targets involved in macropinocytosis in patient brain samples. Data are expressed as mean ± SEM of mRNA intensities of each probe expressed in log 2 ratio. NB: normal brain; grade II, III and IV: glioma samples according to the histopathological grade of the tumor. (**a**) epidermal growth factor receptor (*EGFR*), (**b**) platelet-derived growth factor receptor A (*PDGFRA*) and *PDGFRB*, (**c**) *GIT1*, (**d**) *ARF6*, (**e**) *RAB20*, (**f**) *RAB34*, (**g**) *SWAP70* and (**h**) *SNX5*. Note that only probes with the highest intensities are presented for clarity of the figure with respect to *EGFR* and *SNX5*. Statistical comparisons with the non-tumoral brain samples are represented by (*), statistical comparisons between grade II and IV by (¤), between grade III and IV by (#) and between grade II and III by (§). All statistics are based on two-tailed tests according to conventional thresholds: *p* < 0.05 (*, ¤, # or §), *p* < 0.01 (**, ¤¤, ## or §§) and *p* < 0.001 (***, ¤¤¤, ### or §§§).

**Figure 2 cancers-11-00411-f002:**
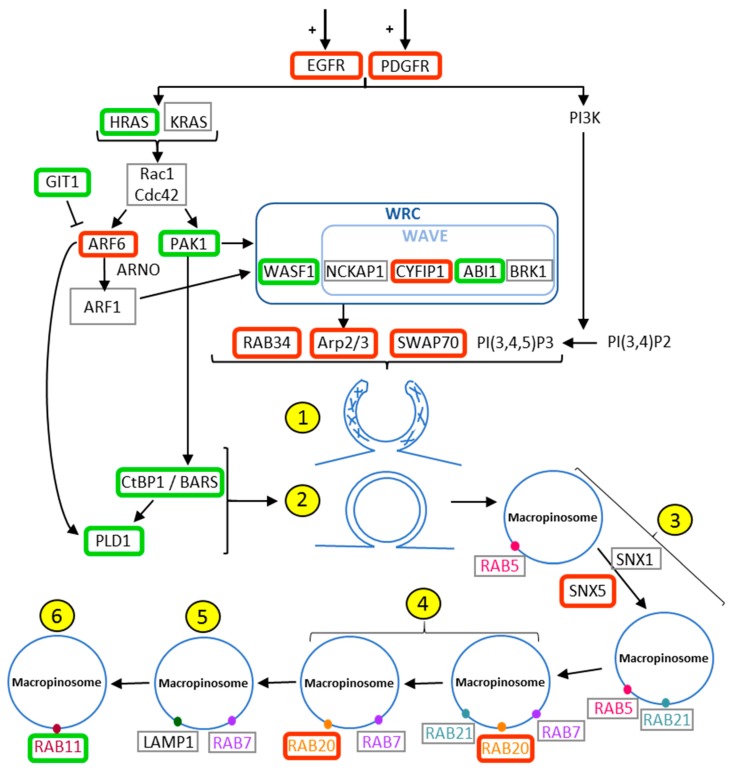
Macropinocytosis process and its deregulation in glioma. Upregulated genes are framed in red, downregulated ones in green, and those genes for which no change was observed are framed in grey. The stimulation of EGFR and PDGFR induces the activation of K-Ras and H-Ras. These Ras GTPases activate both Arf6 and Pak1 to initiate the actin polymerization. Arf6 recruits ARNO (Arf guanine nucleotide exchange factor) for the activation of Arf1 to enable the WAVE regulatory complex (WRC), composed by Wasf1 and the WAVE complex, which regroups Nckap1, Brk1, Cyfip and Abi1. WRC activates Arp2/3, which induces actin branch formation and actin polymerization (1) for the formation of macropinosomes with the help of Rab34, Swap70 and PI(3,4,5)P3. Both Ctbp1 (activated by Pak1) and PLD1 (activated by Arf6 and Ctbp1) are essential for the closure of the macropinocytic cup and the final fission from the plasma membrane (2). Rab5 is a marker of early endosome as well as Rab21, which remains on the intermediate endosome (3). SNX1 and SNX5 play a role in macropinosome maturation (3). Markers of intermediate endosomes are Rab21, Rab20 and Rab7 (4). Rab7 is also a marker of late endosome as well as Lamp1 (5). Rab11 is a marker of endosome recirculating to the plasma membrane (6).

**Figure 3 cancers-11-00411-f003:**
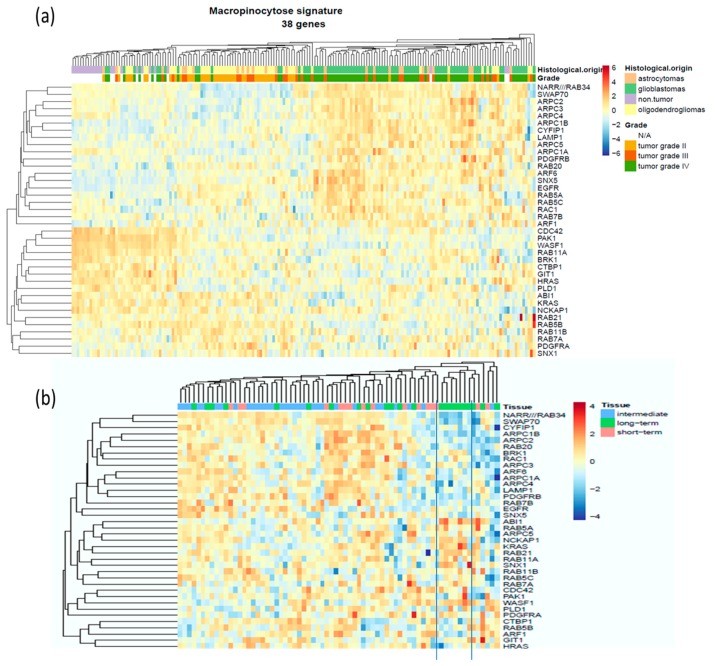
Heatmaps of 38 macropinocytosis gene signatures. (**a**) Dataset GSE4290 regrouping 180 samples (columns) of non-tumoral tissues (N/A) or glial tumors of various histological origin and grade according to the legend. For each gene, the data of the different probes have been averaged to be presented as a single line. Clustering has been made on the basis of the Euclidian distance. (**b**) Similar heatmap made on the dataset from GSE53733 regrouping 70 samples of GBM patients with different overall survival according to the legend. Overexpressed genes are in red, and downregulated genes are in blue (see scale next to the figures).

**Figure 4 cancers-11-00411-f004:**
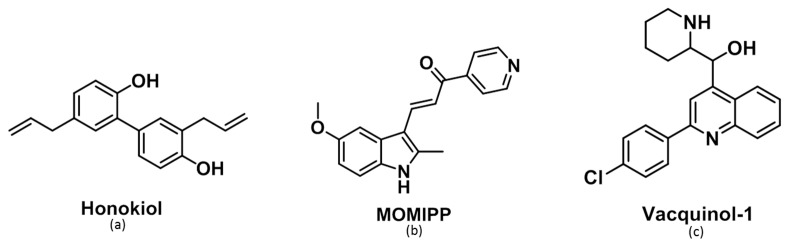
Chemical structure of the three compounds used in this study.

**Figure 5 cancers-11-00411-f005:**
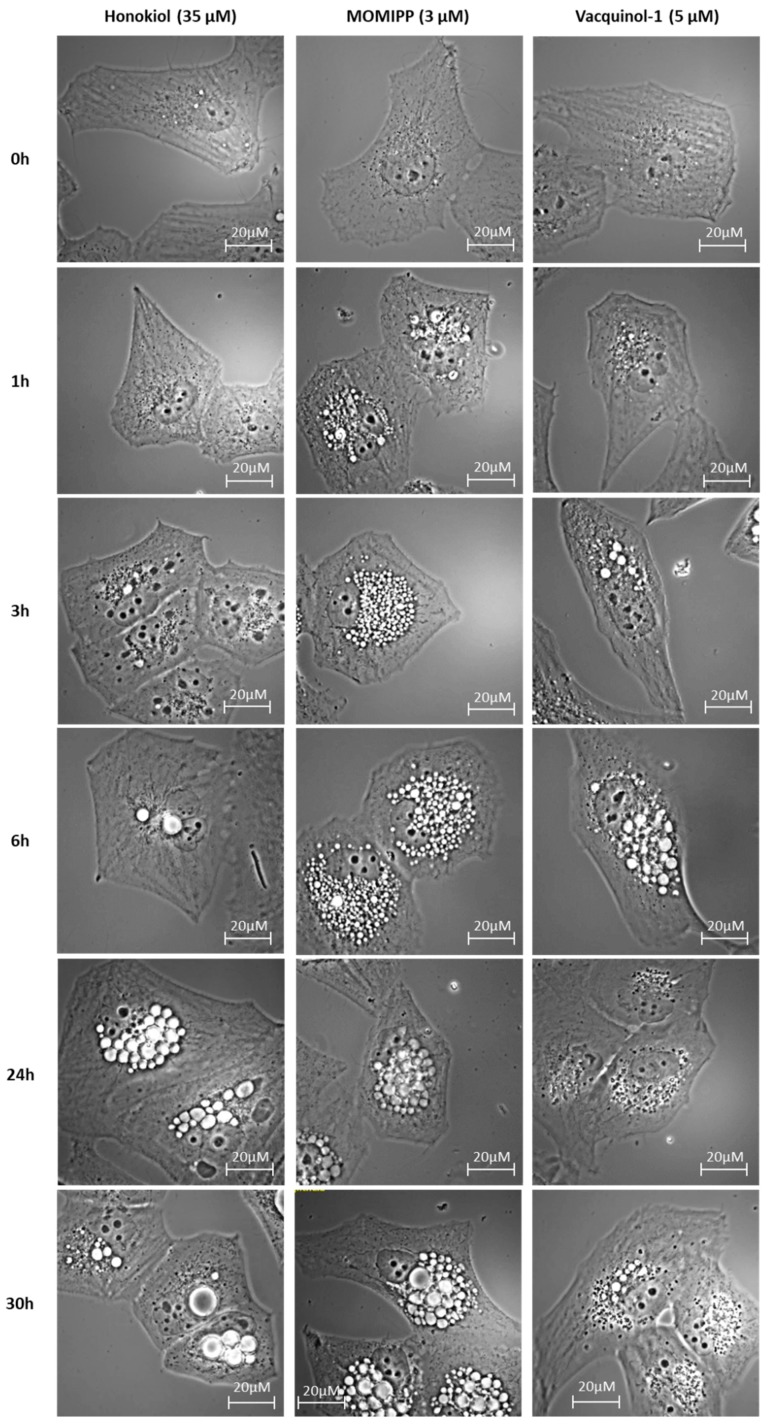
Morphological illustration of the vacuoles formed after treatment of the U373 cell line with honokiol (35 µM), 3-(5-methoxy-2-methyl-1H-indol-3-yl)-1-(4-pyridinyl)-2-propen-1-one (MOMIPP) (3 µM), or vacquinol-1 (5 µM) over time. The accumulation of vacuoles is only visible after 24 h of treatment with honokiol, but those induced by treatment with vacquinol-1 occurs after 3 h and after only 1 h of treatment with MOMIPP. The experiment has been conducted two times in triplicate.

**Figure 6 cancers-11-00411-f006:**
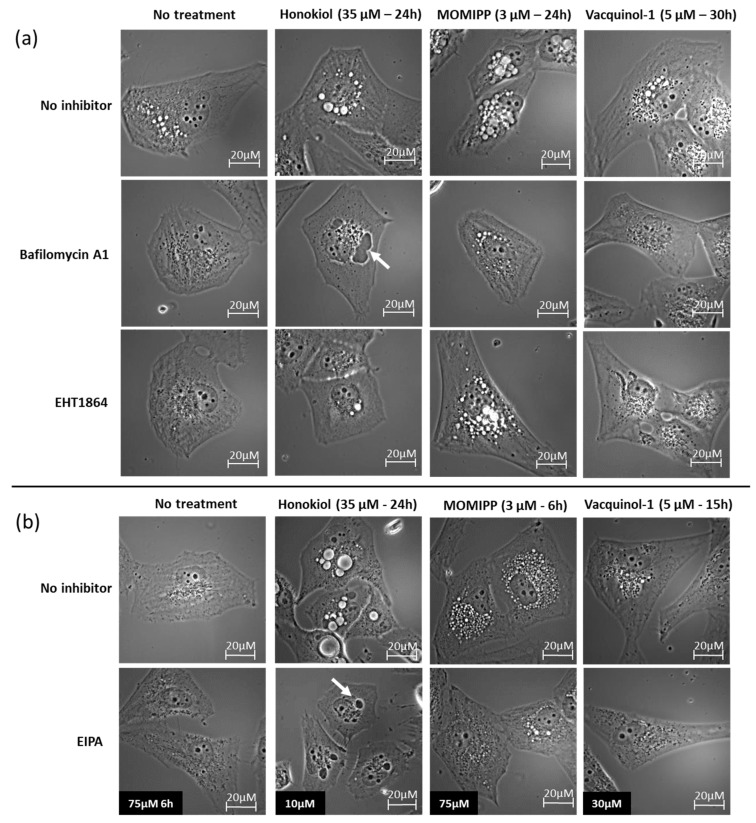
Effects of various inhibitors on vacuole formation following honokiol, vacquinol-1, and MOMIPP treatment in U373 cells. Representative brightfield illustrations of U373 cells treated with or without the inhibitors are as follows: (**a**) pre-treated 1 h with bafilomycin A1 (100 nM) before honokiol (35 µM), MOMIPP (3 µM) or vacquinol-1 (5 µM) treatment, co-treatment with EHT1864 (25 µM) and the compound. Bafilomycin A1 inhibits the vacuoles induced by each compound but not ER-derived dark dense vacuoles induced by honokiol (white arrow). EHT1864 slightly inhibits vacuolization induced by honokiol and vacquinol-1 but not for vacuolization induced by MOMIPP. (**b**) Cells were co-treated with EIPA (10, 30 and 75 µM) and each compound. EIPA inhibits the vacuoles induced by each compound except the ER-derived dark dense vacuoles induced by honokiol (white arrow; similar than bafilomycin A1 effects). Each experiment has been conducted at least twice in triplicate.

**Figure 7 cancers-11-00411-f007:**
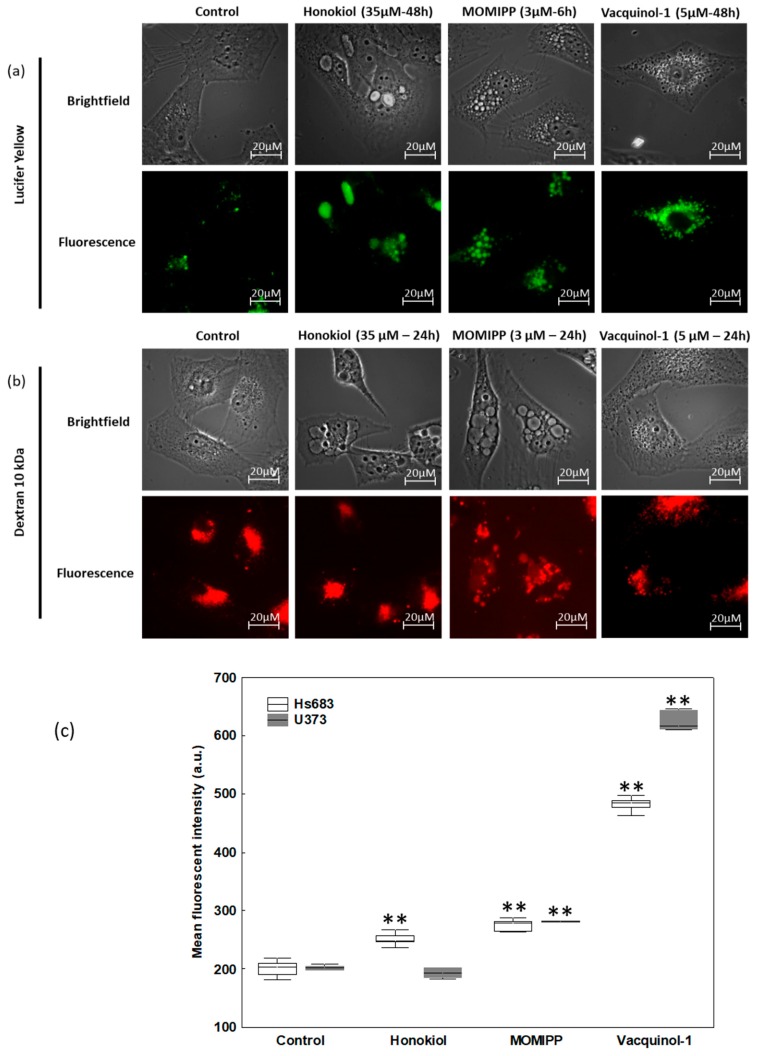
Lucifer yellow and fluorescent 10 kDa dextran uptake by U373 cells treated with honokiol, MOMIPP, or vacquinol-1. (**a**) Illustration of Lucifer yellow staining (100 µg/mL during the whole treatment period). Exposure time is the same for all conditions. The experiment has been conducted three times in duplicate. (**b**) Illustration of fluorescent 10 kDa dextran uptake. Again, cells were treated with each compound in presence of 10 kDa Texas-Red labeled dextran (125 µg/mL) for 24 h. Some vacuoles are positive with MOMIPP and vacquinol-1 but none with honokiol. Exposure time has been adjusted for each condition (550 ms for the control, 490 ms for honokiol, 650 ms for MOMIPP and 160 ms for vacquinol-1). The experiment has been conducted twice in duplicate. (**c**) Quantitative dextran 10 kDa uptake by U373 and HS683 determined by flow cytometry after 24 h and 30 h, respectively. Data are expressed as box plots: line, median; boxes, percentiles 25–75; and whiskers, non-outlier ranges of six experiments. Statistical comparisons with untreated cells are based on Mann–Whitney tests according to conventional thresholds: *p* < 0.05 (*), *p* < 0.01 (**) and *p* < 0.001 (***).

**Figure 8 cancers-11-00411-f008:**
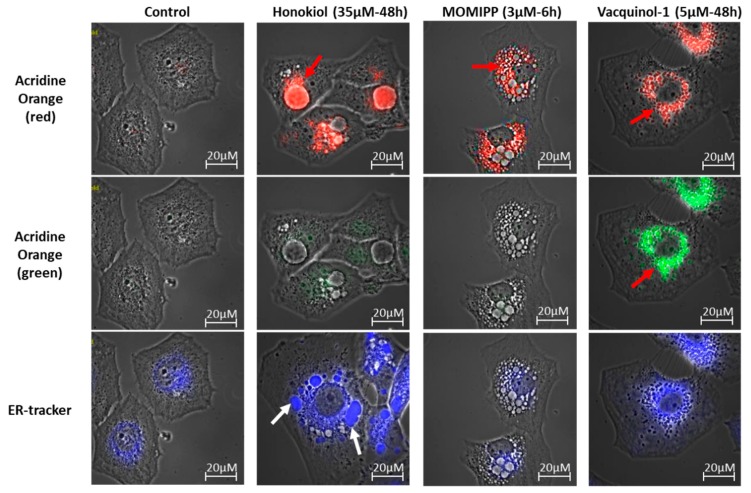
Illustration of the fluorescent characterization of the vacuoles using different fluorescent cell compartment trackers in U373 cells. Two kinds of vacuoles can be observed after treatment with honokiol: red arrows show acridine orange positive vacuoles possibly derived from macropinocytosis, and white arrows show ER-positive vacuoles. Some vacuoles induced by MOMIPP are red-positive with acridine orange (red arrow) but none to the ER-tracker. Vacquinol-1-induced vacuoles are red-positive (red arrow), but also green-positive stained by acridine orange. This experiment has been conducted three times in duplicate.

**Figure 9 cancers-11-00411-f009:**
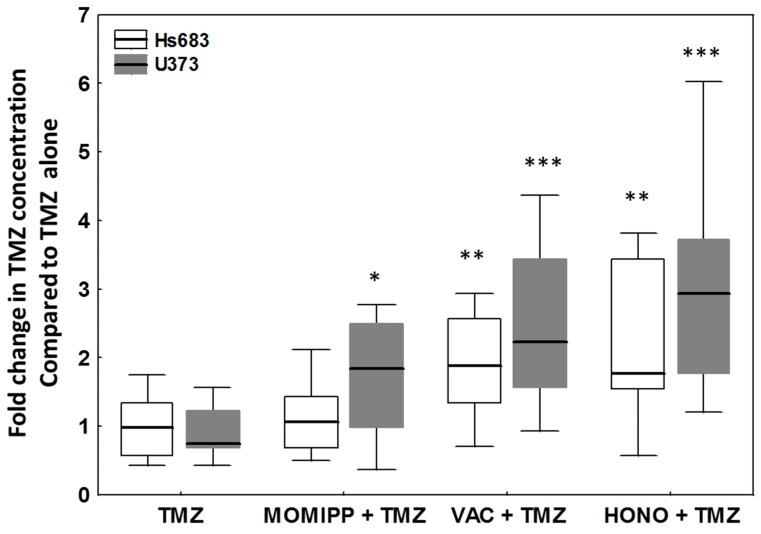
Temozolomide (TMZ) concentrations in U373 and Hs683 cell lysates after two hours of treatment with TMZ (200 µM). Cells were pre-treated with honokiol for 22 h (HONO + TMZ), vacquinol-1 for 15 h (VAC + TMZ) or MOMIPP for 3 h (MOMIPP + TMZ). Data are expressed as fold-change TMZ concentrations compared to its own control (TMZ alone) by box plots: line, median fold change; boxes, percentiles 25–75; whiskers, non-outlier ranges of 12 replicates. TMZ concentration of the cells treated with TMZ alone was normalized to 1. Statistical comparison has been made by the Mann–Whitney test. *p* < 0.05 (*), *p* < 0.01 (**) and *p* < 0.001 (***).

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
