# Peer review of "Dysregulation of Macropinocytosis Processes in Glioblastomas May Be Exploited to Increase Intracellular Anti-Cancer Drug Levels: The Example of Temozolomide"

_cancers, 2019, doi:10.3390/cancers11030411_

Round 1
Reviewer 1 Report
In this manuscript, the authors demonstrated that the expression of the maker genes for macropinocytosis were reduced in patient brain samples. The authors showed that the treatments of three compounds MOMIPP, vacquinol-1 and Honokiol induced large vacuoles in U373 cells over time. Interestingly, they showed that such treatments resulted in an increase in the intake of TMZ, the current first line chemotherapeutic agent for GBM. However, many concerns need to be addressed.
1. This study failed to demonstrate that the vacuoles induced by MOMIPP, vacquinol-1 and Honokiol in GBM cell lines were macropinosomes. Macropinocytosis uptakes large volume of extracellular content. Thus, One of criteria for macropinocytosis is to be able to intake large proteins (e.g., FITC-BSA and DQ-BSA). Second, amiloride and its derivative EIPA are unique inhibitors for macropinosome formation. Third, Src and Ras are critical for the induction of macropinocytosis. Therefore, to demonstrate the vacuoles are macropinosomes, the authors should show that: (i) vacuoles can intake large proteins; and (ii) amiloride treatment as well as inhibition of Src and Ras abolish vacuole formation.
2. Fig 6 showed that Bafilomycin, a lysosome inhibitor, completely abolished vacuole formation. This seems indicate that vacuoles are not macropinosomes.
3. It is unclear whether the increased uptake of TMZ enhances the effect of TMZ against GBM cells.
Author Response
Reviewer #1:
In this manuscript, the authors demonstrated that the expression of the maker genes for micropinocytosis was reduced in patient brain samples. The authors showed that the treatments of three compounds MOMIPP, vacquinol-1 and Honokiol induced large vacuoles in U373 cells over time. Interestingly, they showed that such treatments resulted in an increase in the intake of TMZ, the current first line chemotherapeutic agent for GBM.
ð We thank Reviewer 1 for this positive comment.
However, many concerns need to be addressed.
1. This study failed to demonstrate that the vacuoles induced by MOMIPP, vacquinol-1 and Honokiol in GBM cell lines were macropinosomes. Macropinocytosis uptakes large volume of extracellular content. Thus, One of criteria for macropinocytosis is to be able to intake large proteins (e.g., FITC-BSA and DQ-BSA). Second, amiloride and its derivative EIPA are unique inhibitors for macropinosome formation. Third, Src and Ras are critical for the induction of macropinocytosis. Therefore, to demonstrate the vacuoles are macropinosomes, the authors should show that: (i) vacuoles can intake large proteins; and (ii) amiloride treatment as well as inhibition of Src and Ras abolish vacuole formation.
ð We thank Reviewer 1 for this meaningful comment and those precise suggestions. We have performed accordingly the following experiments:
1) We made use of EIPA and check for vacuolization inhibition. We found that EIPA inhibited vacuolization induced by each of three compounds (with the exception of ER-derived vacuoles observed in honokiol treated of course). Those results are provided in the new Figures 6 and S3.
2) To address the question of large protein intake by macropinocytosis and better differentiate macropinocytosis from other endocytic processes, we did not made use of fluorescent albumin as suggested because it would have possibly altered free drug concentration and/ or drug intracellular concentration. We thus made use of the well- known alternative fluorescent dextrans of high molecular weights, i.e. 10kDa and 70kDa; Galemkamp et al, 2019). Results are provided in Figures 7, S6 and S7and discussed in both Results and Discussion section accordingly.
3) To address the possible role of Src, we evaluated whether PP2, a commercial inhibitor of it affects vacuolization induced by honokiol, MOMIPP and vacquinol-1. We found that 25 µM of PP2 indeed inhibited the apparition of the enlarged vacuoles in honokiol treated cells (results provided in Figure SI 4). At that concentration PP2 was unable to inhibit vacuolizations induced by MOMIPP and vacquinol-1. We tried higher concentrations to possibly inhibit MOMIPP and vacquinol-1 vacuoles but encountered solubility issues making us impossible to conclude at this stage whether Src may or not be involved in the vacuolization processes induced by these two compounds. These informartions and results have been added in the manuscript and the supplementary materials and addressed in the Discussion section.
2. Fig 6 showed that Bafilomycin, a lysosome inhibitor, completely abolished vacuole formation. This seems indicate that vacuoles are not macropinosomes.
ð We totally agree with Reviewer 1 that Bafilomycin A1 is not a specific inhibitor of macropinocytosis. We have stated more clearly in the manuscript the effects Bafilomycin 1 on the downstream events of macropinocytosis pathway according to Recouvreux and Commisso 2017. The following paragraph has been added line 211:
“Bafilomycin A1 is an inhibitor of the vacuolar H+-ATPase that plays crucial roles in maintaining low pH of late endosomes and lysosomes. Bafilomycin A1 was accordingly shown to blocks the endosomal and endosome-lysosomal fusion during macropinocytosis [37]. Additionally, bafilomycin A1 has been suggested to also inhibit nascent macropinosome formation, similarly to Na+/H+ exchanger inhibitor, by disrupting the fine tuning of submembranous pH needed for the recruitment and activation of Rac1 and Cdc42 to membrane ruffles [76] »
ð In addition, we conducted also assay with the EIPA inhibitor, considered as a more specific macropinocytosis inhibitor as mentioned and suggested by Reviewer 1 (point 1 above) (see above).
3. It is unclear whether the increased uptake of TMZ enhances the effect of TMZ against GBM cells.
ð We totally agree that our study does not provide evidence that increased uptake of TMZ is able to enhance its therapeutic effects or not. While previous publication already showed such improvement with respect to honokiol in vitro (reference # 77 of our revised manuscript), similar studies have not been conducted with respect to vacquinol-1 and MOMIPP. Importantly, as clearly stated in our Discussion, TMZ is characterized by an excellent uptake. As emphasized by Reviewer 3, our study actually aims to provide proof of concept that may be applied to other drugs. According to Reviewers 1 and 3, we addressed their comments in the new version of our manuscript in the Discussion section. We also adapted our title that may better reflect the concept proposed in our study (as requested by Reviewer 2).

Reviewer 2 Report
1. Title
The title can not reflect main findings of this work. And the authors should present a new title for the paper, which many summarize the main findings of their work..
2. Abstract
The abstract is ok.
.
3. Key words
Ok.
4. Introduction
Acceptable.
5. Materials and Methods
Ok.
When the reagent manufacture appears the second time, it is not necessary to present its detailed information.
6. Statistical analysis
Ok.
7. Results and figure legends
Line 131-133 The authors should consider rewriting the sentence “Even thought the mRNA expressionanalysis may be of limited value when considering the importance of protein expression level and activation, we were surprised by the number of targets whose mRNA levels are deregulated.”
8. Discussion
Ok based on the present results.
One paper has recently been published in Cancers to elucidate the mechanisms under which honokiol inhibits glioma stem cell-like cells. This may facilitate discussion of honokiol roles in this paper.
Fan YP, Xue WK, Schachner M, Zhao WJ. Honokiol eliminates glioma/glioblastoma stem cell-like cells via JAK-STAT3 signaling and inhibits tumor progression by targeting epidermal growth factor receptor. Cancers, 2019, 11(1), 22; https://doi.org/10.3390/cancers11010022
Line 335-336
The authors should rewrite the sentence “Nevertheless, we highlighted that important key actors of macropinocytosis appeared obviously deregulated in aggressive gliomas and GBM in particular.”.
Line 390
The authors should rewrite the sentence “Perspectives include evaluation---".
Line 451
“2µM” is not standard. Please check and correct similar errors.
9. Figures and tables
The reviewer detected some minor grammatical errors, which the authors should check carefully and correct.
For example, “Upregulated genes are frame ---” in line 141.
The authors should modify the size of the red arrows in figure 7.
Please present the figures in the order that their corresponding descriptions appear. For example, figure 2 should be reordered.
Author Response
Please find below our point by point response to your comments and requests
Answers are highlighted in blue italic font
Reviewer #2:
1. Title
The title can not reflect main findings of this work. And the authors should present a new title for the paper, which many summarize the main findings of their work.
ð We propose a new title addressing also comments of Reviewers 1 and 3 and hope that now, the title better reflects the findings described in our study. The new title proposed is “Dysregulation of macropinocytosis process in glioblastomas may be exploited to increase intracellular anti-cancer drug: the example of temozolomide.”
2. Abstract
The abstract is ok.
ð We thank Reviewer 2 for this positive comment
3. Key words
Ok.
ð We thank Reviewer 2 for this positive comment
4. Introduction
Acceptable.
ð We thank Reviewer 2 for this positive comment
5. Materials and Methods
Ok.
ð We thank Reviewer 2 for this positive comment
When the reagent manufacture appears the second time, it is not necessary to present its detailed information.
ð We have checked for repetition and remove them
6. Statistical analysis
Ok.
ð We thank Reviewer 2 for this positive comment
7. Results and figure legends
Line 131-133 The authors should consider rewriting the sentence “Even though the mRNA expression analysis may be of limited value when considering the importance of protein expression level and activation, we were surprised by the number of targets whose mRNA levels are deregulated.”
ð We actually addressed this request together with the other request raised at point 9 below by the same Reviewer consisting in re-ordering the figures, e.g. figure 2. This sentence was thus removed from its original place and the info has been used to introduce at the right place figure 2, providing a better link between new Figures 1 and 3 at lines 140-144 of the track change version of our revised manuscript.
8. Discussion
Ok based on the present results.
ð We thank Reviewer 2 for this positive comment
One paper has recently been published in Cancers to elucidate the mechanisms under which honokiol inhibits glioma stem cell-like cells. This may facilitate discussion of honokiol roles in this paper.
Fan YP, Xue WK, Schachner M, Zhao WJ. Honokiol eliminates glioma/glioblastoma stem cell-like cells via JAK-STAT3 signaling and inhibits tumor progression by targeting epidermal growth factor receptor. Cancers, 2019, 11(1), 22; https://doi.org/10.3390/cancers11010022
ð We thank Reviewer 2 for this suggestion and have made used of that article to enrich our discussion lines 506-508
Line 335-336
The authors should rewrite the sentence “Nevertheless, we highlighted that important key actors of macropinocytosis appeared obviously deregulated in aggressive gliomas and GBM in particular.”.
ð We have rewritten this sentence. You can now find lines 420-426 of the track change version of our revised manuscript.
“Nevertheless, as highlighted by Figure 2, macropinocytosis process appeared obviously deregulated in glioma and particularly in GBM (Fig. 1). Accordingly, we showed that the mRNA expression signature of these 38 genes taken together and covering most of the macropinosome formation, maturation, and turn-over processes enabled discriminating GBM from non-tumoral samples and lower grade glioma on the basis of unsupervised analysis (Fig. 3) »
Line 390
The authors should rewrite the sentence “Perspectives include evaluation---".
ð The perspective paragraph has been revised and rewritten according to the other changes made in the Discussion (consequently to the new experiments and to Reviewers 1 and 3 comments)
Line 451
“2µM” is not standard. Please check and correct similar errors.
ð The correction has been made.
9. Figures and tables
The reviewer detected some minor grammatical errors, which the authors should check carefully and correct.
For example, “Upregulated genes are frame ---” in line 141.
ð The corrections have been made.
The authors should modify the size of the red arrows in figure 7.
ð The size of the arrows has been reduced.
Please present the figures in the order that their corresponding descriptions appear. For example, figure 2 should be reordered.
ð Figures have been re-ordered (original figures 1 and 2 moved according to the text; note that new figures have been added too).
Reviewer 3 Report
This is an interesting manuscript detailing the potential impact of macropinocytosis in glioma cells. The authors clearly review this process and subsequent trafficking to the lysosome, and they provide nice data linking deregulation of this process to increased intracellular accumulation of temozolomide chemotherapy. While the differences in TMZ accumulation may not particularly translate to profound efficacy effects, the impact of this process on internalization of therapeutic antibodies and/or antibody drug conjugated could be much more relevant. While additional experimentation is not necessarily required, the authors should discuss their findings in the context of pre-clinical and/or clinical results in GBM with Depatux-M (ABT-414), which has shown some promising activity in GBM. Otherwise, there are only minor issues that should be addressed.
Unclear from legend what the significance of the *** notation vs * in Figure 3
Figure 5 should provide images from untreated – time 0 – samples.
Figure 8 should provide statistical comparisons between TMZ alone and each of the co-treatments.
In Figures 5-7, the number of independent experiments performed should be noted in the legend.
Author Response
Please find below our point by point response to your comments and requests
Answers are highlighted in blue italic font
Reviewer #3:
This is an interesting manuscript detailing the potential impact of macropinocytosis in glioma cells. The authors clearly review this process and subsequent trafficking to the lysosome, and they provide nice data linking deregulation of this process to increased intracellular accumulation of temozolomide chemotherapy.
ð We thank Reviewer 1 for this positive comment.
While the differences in TMZ accumulation may not particularly translate to profound efficacy effects, the impact of this process on internalization of therapeutic antibodies and/or antibody drug conjugated could be much more relevant. While additional experimentation is not necessarily required, the authors should discuss their findings in the context of pre-clinical and/or clinical results in GBM with Depatux-M (ABT-414), which has shown some promising activity in GBM.
ð We totally agree with Reviewer 3 comment. Reviewer 1 also pointed the question about the impact in terms of efficacy of TMZ (point 3 of Reviewer 1). Indeed our study does not provide evidence that increased uptake of TMZ is able to enhance its therapeutic effects or not. While previous publication already showed such improvement with respect to honokiol in vitro (reference # 77 of our revised manuscript), similar studies have not been conducted with respect to vacquinol-1 and MOMIPP. As emphasized here by Reviewer 3, our study actually aims to provide proof of concept that may be applied to other drugs. We addressed Reviewers 1 and 3 comments in the new version of our manuscript in the Discussion section. We have used the pertinent example of depatuxizumab mafodotin provided by Reviewer 3 to rewrite part of our Discussion (lines 485-493 of the track change version of our revised manuscript).
ð We also adapted our title that may better reflect the concept proposed in our study (as requested by Reviewer 2).
Otherwise, there are only minor issues that should be addressed.
Unclear from legend what the significance of the *** notation vs * in Figure 3
ð Detailed legends for statistical signs used are now provided into the legends of each figure
Figure 5 should provide images from untreated – time 0 – samples.
ð Pictures at t=0 have been added. Note that the figure had to be rotated to portrait orientation.
Figure 8 should provide statistical comparisons between TMZ alone and each of the co-treatments.
ð We have conducted additionnal data sets to allow statistical comparison. Mann Whitney comparison has been performed and confirmed significant increase in TMZ uptake with the combined treatments (see new Figure 9).
In Figures 5-7, the number of independent experiments performed should be noted in the legend.
ð Each legend details now the number of independent experiments conducted and the number of replicates in each of them.
Round 2
Reviewer 1 Report
no comment